# Native Bovine Hydroxyapatite Powder, Demineralised Bone Matrix Powder, and Purified Bone Collagen Membranes Are Efficient in Repair of Critical-Sized Rat Calvarial Defects

**DOI:** 10.3390/ma13153393

**Published:** 2020-07-31

**Authors:** Alexey Veremeev, Roman Bolgarin, Vladimir Nesterenko, Alexander Andreev-Andrievskiy, Anton Kutikhin

**Affiliations:** 1Matriflex LLC, 12 Aviakonstruktora Mikoyana Street, Building A, 2nd Floor, Office 1, Moscow 125252, Russia; alexey.veremeev@nearmedic.ru (A.V.); roman.bolgarin@nearmedic.ru (R.B.); 2Department of Immunology, Gamaleya National Research Centre of Epidemiology and Microbiology, 18 Gamaleya Street, Moscow 123098, Russia; vladimir.nesterenko@nearmedic.ru; 3Faculty of Biology, Moscow State University, 73A, Leninskie Gory Street, Moscow 119330, Russia; aandrievsky@gmail.com; 4Laboratory for Vascular Biology, Department of Experimental Medicine, Research Institute for Complex Issues of Cardiovascular Diseases, 6, Sosnovy Boulevard, Kemerovo 650002, Russia

**Keywords:** bone defects, bone repair, bone autografts, xenogeneic implants, hydroxyapatite, demineralised bone matrix, bone collagen, critical-sized rat calvarial defect, microcomputed tomography, clinical translation

## Abstract

Here we evaluated the efficacy of bone repair using various native bovine biomaterials (refined hydroxyapatite (HA), demineralised bone matrix (DBM), and purified bone collagen (COLL)) as compared with commercially available bone mineral and bone autografts. We employed a conventional critical-sized (8 mm diameter) rat calvarial defect model (6-month-old male Sprague–Dawley rats, *n* = 72 in total). The artificial defect was repaired using HA, DBM, COLL, commercially available bone mineral powder, bone calvarial autograft, or remained unfilled (*n* = 12 animals per group). Rats were euthanised 4 or 12 weeks postimplantation (*n* = 6 per time point) with the subsequent examination to assess the extent, volume, area, and mineral density of the repaired tissue by means of microcomputed tomography and hematoxylin and eosin staining. Bovine HA and DBM powder exhibited excellent repair capability similar to the autografts and commercially available bone mineral powder while COLL showed higher bone repair rate. We suggest that HA and DBM powder obtained from bovine bone tissue can be equally applied for the repair of bone defects and demonstrate sufficient potential to be implemented into clinical studies.

## 1. Introduction

Despite the recent improvements in medical imaging, implementation of novel treatment approaches, and spread of current advances in orthopaedic care, increase in efficiency and rehabilitation after the orthopaedic surgery remains a major goal of contemporary medicine [1]. This is particularly important for working-age adults because of a high incidence of injuries and bone diseases in this population cohort [2,3,4,5]. In certain patient groups, disability rates remain unacceptably high even in spite of timely and proper diagnostics and treatment [2,3,4,5]. Therefore, harnessing of bone regeneration by using xenogeneic bone implants instead of ceramic or metal alternatives is still an unmet clinical need [6,7,8].

The use of bone implants is now established as the gold standard in bone replacement and filling of the bone defects after severe injuries requiring extensive surgical interventions [9,10,11]. Albeit autografts are the option of choice [9], their use is also associated with major bleedings, infections, and chronic postoperative pain and is restricted by the limited sources of appropriate autologous bone material [10,11]. Allografts have significantly lower osteoconductive and osteoinductive potential and also may cause infections and immune rejection [10,11]. Efficient bone repair can be potentially reached through the tissue engineering approaches that combine filling of the defect with enhancing proliferation and matrix production by the host osteoblasts [12,13,14,15,16]. In this treatment modality, xenogeneic bone implants are used as fillers due to their excellent biocompatibility and superior mechanical characteristics.

Bone hydroxyapatite (HA, Ca_10_(PO_4_)_6_(OH)_2_) and type I collagen are responsible for biophysical and functional features of the bone tissue [17,18]. Extraction of the bone mineral from the bulk tissue is relatively easy and can be standardised to obtain a biomaterial of perfect purity [19]. Trabecular or porous structure of native HA ensures its high permeability for therapeutic agents and cell populations, which is necessary for successful targeted therapy [20,21,22]. Importantly, HA can be produced in a segmental form to prepare scaffolds [20], as micro- or nanosized powder to completely fill the defect [21,22], or incorporated into the polymer composites as a supplement for increasing osteoconductive and osteoinductive features [23,24]. Another promising xenogeneic filler is bone extracellular matrix, which is 85–90% composed of type I collagen and contains multiple growth factors, also being highly permeable for therapeutic agents and cells because of fibrous network and numerous pores [17,18,25,26,27]. Type I collagen is notable for high structure similarity between species and low immunogenicity due to its evolutionary conservativeness [28] and can also be manufactured in different dosage forms permitting its separate use where appropriate [29,30,31,32,33].

We have recently developed a number of original protocols for extracting native bovine refined HA powder, demineralised bone matrix powder (DBM), and purified bone collagen membranes (COLL), which can be potentially applied in orthopaedic surgery and dentistry. Here we evaluated the efficacy of these bovine-derived bone substitutes in the repair of critical-sized rat calvarial defects.

## 2. Results

### 2.1. Filling of Critical-Sized Calvarial Defect with Bovine HA, DBM, and COLL Is Safe for Experimental Animals

To examine the efficacy of various bone substitutes for bone repair, we applied a widely established critical-sized (8 mm diameter) rat calvarial defect model [34]. The justification behind the use of this model was that the defect of the indicated size is unable to be spontaneously repaired, therefore requiring implantation of the bone tissue to be filled [34]. Four or twelve weeks postoperation, animals were euthanised with the subsequent excision of calvarial bones and adjacent tissues at the operation site. We further conducted histological and microcomputed tomography examination to evaluate the volume and area of the repaired tissue and calculate the tissue and bone mineral density. Experimental design is summarised in Table 1.

None of the rats showed a significant decrease in body weight in the early postoperative period (Appendix A) and during the entire observation period (Appendix A) regardless of the implant used. Most important, body weight did not change significantly (≥15%) across the experimental groups at any time point (Appendix A) testifying to the general safety of the implants for experimental animals.

Complete blood count measured at 12 weeks postimplantation also did not show any statistically differences across the groups in terms of white blood cell count (Appendix A), red blood cell count (Appendix A), haemoglobin (Appendix A), haematocrit (Appendix A), mean corpuscular volume (Appendix A) measurements, and platelet count (Appendix A).

In agreement with the previous findings, biochemical analysis did not reveal statistically significant differences regarding any of the measured parameters (Appendix A). Collectively, these results indicated sufficient safety of the investigated bone fillers.

### 2.2. Bovine HA and DBM Are Highly and Equally Efficient for the Repair of Critical-Sized Rat Calvarial Defect While COLL Demonstrates the Highest Bone Repair Rate

Microcomputed tomography examination confirmed the successful implantation (Figure 1A) and showed complete repair of the defect upon its filling with autograft, reference product (Geistlich Bio-Oss^®^), HA, or DBM as early as 4 weeks postoperation (Figure 1B) whilst COLL showed major repair efficiency only 12 weeks postimplantation (Figure 1C). Unfilled defect did not display any repair signs testifying to the relevance of used implantation model (Figure 1B,C).

Hematoxylin and eosin staining verified the results of microcomputed tomography examination, indicating comparable repair capability of the autograft, reference product (Geistlich Bio-Oss^®^), HA, DBM, and COLL both after 4 (Figure 2A) and 12 weeks postimplantation (Figure 2B), albeit the latter implant was notable for the different morphology of the repaired tissue (Figure 2A,B).

Semi-quantitative analysis of the microcomputed tomography images revealed the highest volume of the repaired tissue upon the implantation of HA, while autografts, the reference product (Geistlich Bio-Oss^®^), and DBM showed lower values; however, these differences did not reach the statistical significance (Figure 3A). COLL demonstrated rapid bone repair rate from the 4th to the 12th weeks postimplantation (Figure 3A). Similar results were obtained when measuring bone tissue area, yet HA was equally efficient to autografts, reference product (Geistlich Bio-Oss^®^), and DBM in this case (Figure 3B). Mineral density of the repaired bone and surrounding tissue was the highest in autografts in comparison to other implants (Figure 3C) whereas bone mineral density did not differ significantly across the experimental groups (Figure 3D). Taken together, these results suggested excellent bone repair capability of the tested implants, in particular HA and DBM.

## 3. Discussion

According to the Global Burden of Disease estimates, injuries are responsible for 1 billion of admissions and 5 million deaths annually [3]. Annually, surgeons carry out around 4 million of bone grafting interventions [35], the majority of which (1.5 million) are performed in the United States [36]. Therefore, there is a large and urgent clinical need in commercially available, off-the-shelf, ready-to-use implants for bone grafting [35,36]. According to various estimates, annual shortage of bone grafts in Russian Federation reaches 150,000–275,000 implants. Despite some excellent solutions in this field are established in the market (e.g., Geistlich Bio-Oss^®^, a native bone mineral powder), their cost-effectiveness in developing countries remains vague.

Hence, our group developed the original protocols to extract bovine HA (powder), DBM (powder), and COLL (membranes). The mentioned techniques include sequential mechanical removal of the soft tissues, disinfection, deproteinisation (or removal of contaminating proteins), decellularisation, delipidation, DNase treatment, optional demineralisation, drying, and fractionation of biomaterial with the subsequent sterilisation by ethylene oxide. Obtaining of COLL also requires purification, salt precipitation, and enrichment of collagen with the subsequent lyophilisation and sterilisation.

This study demonstrated that filling of critical-sized calvarial defect with HA or DBM was safe for the experimental animals and led to the complete repair of the defect similar to the reference bone mineral powder and autografts, whereas COLL exhibited an excellent bone repair rate. These results indicated high osteoconductive and osteoinductive properties of the extracted xenogeneic bone fillers.

In accordance with our findings, ovine bone mineral showed better integration with rabbit femoral bones and enhanced vascularisation in comparison with artificially synthesised HA and β-tricalcium phosphate [20]. Further, bovine HA accelerated bone regeneration in patients who underwent sinus lift surgery in contrast to synthetic β-tricalcium phosphate [22]. This may be explained by the induction of osteogenic differentiation specific for native HA [37]. DBM has been proposed as an another option to stimulate bone regeneration, which can be as efficient as native HA or autografts in a rabbit spinal fusion model [38] and filling of rat radial bone defects [39] or canine tibial defects [40]. Addition of bone collagen to the native bone mineral further improves osteoconductivity and regeneration of calvarial bones in rabbits [41], also promoting adhesion, proliferation and osteogenic differentiation of murine preosteoblasts in comparison with a native bone mineral [32].

To conclude, bovine HA and DBM have pronounced repair capability in a critical-sized rat calvarial defect model. COLL is characterised by the highest bone repair rate among all tested implants. We suggest that HA and DBM extracted from the bovine bone tissue can be equally applied for the repair of bone defects and show sufficient potential to be implemented into clinical studies.

## 4. Materials and Methods

### 4.1. Implant Preparation

Bovine femoral bones were obtained from the meat processing plant (Mikkon, Konoplino, Russian Federation) according to the ISO 22442 “Medical devices utilizing animal tissues”. Refined bovine HA powder (0.25–1 mm diameter) with preserved native structure was extracted from the bones according to the following protocol: (1) triple freezing (−40 °C, MDF-U5411, Sanyo, Osaka, Japan) and thawing; (2) sawing, removal of the soft tissues upon heating to 100 °C for 30 min, and disinfection by incubation in 5% H_2_O_2_ (216763, Sigma-Aldrich, St. Louis, MO, USA) for 24 h; (3) triple washing with double distilled water at 4 °C for 2 h; (4) drying at 100 °C (MF-8A, Gilson, Lewis Center, OH, USA) for 24 h; (5) sawing and three rounds of incubation in NaOH (0.1 mmol/L, 221465, Sigma-Aldrich, St. Louis, MO, USA) for 24 h; (6) triple washing with double distilled water at 4 °C for 24 h; (7) drying at 80 °C during 24 h; (8) incubation in 14% ethylenediamine (03550, Sigma-Aldrich, St. Louis, MO, USA) at 120 °C for 48 h; (9) triple washing with double distilled water at 4 °C for 24 h; (10) drying at 100 °C for 24 h; (11) grinding (IKA MF10, Sigma-Aldrich, St. Louis, MO, USA) into 2 mm diameter powder; (12) incubation in 5% ethylenediamine at 120 °C for 48 h; (13) triple washing with double distilled water at 4 °C for 24 h; (14) incubation in DNase (500 µg/mL, DN25, Sigma-Aldrich, St. Louis, MO, USA) for 24 h; (15) triple washing with double distilled water at 4 °C for 24 h; (16) drying at 160 °C for 24 h; (17) drying at 400 °C (MF-8A) for 24 h; (18) fractionation by passing the bone powder through the 1 mm sieve (Z645273, Sigma-Aldrich, St. Louis, MO, USA); (19) removal of the smallest fraction by passing through the 0.25 mm sieve (Z645265, Sigma-Aldrich, St. Louis, MO, USA); and (20) sterilisation by ethylene oxide (Steripack Service, Moscow, Russian Federation).

The protocol for the extraction of bovine DBM powder included: (1) triple freezing (−40 °C) and thawing; (2) sawing, removal of the soft tissues, and disinfection by incubation in 5% H_2_O_2_ for 24 h; (3) triple washing with double distilled water at 4 °C for 2 h; (4) incubation in 1% sodium dodecyl sulphate (436143, Sigma-Aldrich, St. Louis, MO, USA) for 72 h; (5) triple washing with double distilled water at 4 °C for 24 h; (6) incubation in 1% Triton X-100 (T9284, Sigma-Aldrich, St. Louis, MO, USA) for 72 h; (7) triple washing with double distilled water at 4 °C for 24 h; (8) incubation in 95% ethanol at 4 °C for 24 h; (9) triple washing with double distilled water at 4 °C for 24 h; (10) three rounds of incubation in NaOH (0.1 mmol/L) for 24 h; (11) triple washing with double distilled water at 4 °C for 24 h; (12) incubation in DNase (500 µg/mL) for 24 h; (13) triple washing with double distilled water at 4 °C for 24 h; (14) grinding into 2 mm diameter powder; (15) incubation in HCl (0.5 mmol/L, 320331, Sigma) and 1% Triton X-100 for 24 h with the following neutralisation in NaOH (0.5 mmol/L) for 24 h; (16) triple washing with double distilled water at 4 °C for 24 h; (17) drying at 80 °C during 24 h; and (18) sterilisation by ethylene oxide.

COLL (bovine bone collagen) was extracted and purified as follows: (1) triple freezing (−40 °C) and thawing; (2) sawing, removal of the soft tissues, and disinfection by incubation in 5% H_2_O_2_ for 24 h; (3) triple washing with double distilled water at 4 °C for 2 h; (4) three rounds of incubation in NaOH (0.1 mmol/L) for 24 h; (5) triple washing with double distilled water at 4 °C for 24 h; (6) incubation in 95% ethanol at 4 °C for 24 h; (7) triple washing with double distilled water at 4 °C for 24 h; (8) three rounds of incubation in 3% ice-cold acetic acid (695092, Sigma-Aldrich, St. Louis, MO, USA) and pepsin (1 g/L, P6887, Sigma-Aldrich, St. Louis, MO, USA) at 4 °C for 48 h; (9) three rounds of filtration through 100 µm-filters (NY1H09000, Sigma-Aldrich, St. Louis, MO, USA); (10) precipitation of collagen by adding NaCl (56 g/L, S9888, Sigma-Aldrich, St. Louis, MO, USA) for 24 h at 4 °C; (11) filtration of precipitate through 100 µm-filters; (12) dissolution of the precipitate in 1% ice-cold acetic acid containing NaCl (345 g/L) and DNAse (500 µg/mL) and incubation for 24 h; (13) filtration of precipitate through 100 µm-filters; (14) dissolution in in 1% ice-cold acetic acid and lyophilisation (Labconco, Kansas, MO, USA) in the membrane form at −80 °C and 100 Pa during 48 h; and (15) sterilisation by ethylene oxide.

All aforementioned procedures were performed strictly adhering to the ISO 22442 “Medical devices utilizing animal tissues” and ISO 13485 “Medical devices”. Geistlich Bio-Oss^®^ (Geistlich Pharma, Wolhusen, Switzerland), a commercially available native bovine HA powder (0.25–1 mm diameter) sterilised by γ-irradiation, was used as a reference product. Autograft (rat calvarial bones) isolated and reimplanted during the surgery was used as a positive control whereas unfilled bone defect was compared as a negative control.

### 4.2. Animal Model

All animal experiments were performed in Research Institute of Mitoengineering at Moscow State University (Moscow, Russian Federation). The study included 72 male, 6-month-old Sprague–Dawley rats, which were provided by the Core Facility of Research Institute of Bioorganic Chemistry of the Russian Academy of Sciences (Puschino, Russian Federation). After the transport to the Research Institute of Mitoengineering at Moscow State University (Moscow, Russian Federation), rats underwent a 14-day quarantine for the adaptation purposes.

Animals were allocated to the polypropylene cages (780 cm^2^ area, 2 rats per cage, T3, Tecniplast, Buguggiate, Italy) lined with wood chips (Lignocel, J. Rettenmaier and Söhne, Rosenberg, Germany) and had access to the sterilised water and food (rat chow, Chara, Assortiment-Agro, Russian Federation) ad libitum. Throughout the whole time of experiment, the standard conditions of the temperature (23 ± 3 °C), relative humidity (50% ± 20%), and 12 h light/dark cycles were carefully maintained, and the health status of all rats was monitored daily. Randomisation to allocate animals to experimental groups was performed using GraphPad Prism 8 (GraphPad Software, San Diego, CA, USA). There were no specific inclusion or exclusion criteria. Experiments were performed in a blinded fashion. After the surgery, all animals were individually marked by ear punching and all cages were additionally marked by the paper cards for the identification purposes. Rats were weighed (Pioneer PA2102, Mettler Toledo, Columbus, OH, USA) daily during the first week upon the surgery and then weekly until the end of the experiment. All procedures were carried out in conform with the European Convention for the Protection of Vertebrate Animals used for Experimental and other Scientific Purposes and have been approved by the Ethical Committee of the Research Institute for Mitoengineering at Moscow State University (Moscow, Russian Federation) (ethical approval code 37/2019, approved on 17 June 2019).

To evaluate the repair efficiency of the abovementioned bone implants, we employed widely established critical-sized (8 mm diameter) rat calvarial defect model [34]. Induction of anaesthesia was performed by an intraperitoneal injection of tiletamine (20 mg/kg), zolazepam (20 mg/kg), and xylazine (6 mg/kg). After the shaving of the area from the bridge of the snout between the eyes to the caudal end of the calvarium and disinfection of the operative site by 70% ethanol, we made an approximately 1.5 cm incision down to periosteum over the scalp from the nasal bone to caudal to the middle sagittal crest. The periosteum down the sagittal midline was then divided by the scalpel (Figure 4A). After spreading of the soft tissues and exposing the underlying bone, calvarium was scored with the surgical drill and 8 mm diameter trephine operating at 1000 rpm, with a regular irrigation with a sterile 0.9% NaCl solution (Figure 4B). The excised bone fragment (Figure 4C) was then removed using an elevator (Figure 4D). Upon checking the integrity of dura mater and blood vessels, the defect was washed with a sterile 0.9% NaCl solution to remove any debris and bone chips. The defect was then filled (Figure 4E) with bovine HA, DBM, COLL, Geistlich Bio-Oss^®^, autograft (excised calvarial bones, positive control), or remained unfilled (negative control). Periosteum over the implant and skin over the periosteum were closed (Figure 4F) by 4-0 Monocryl monofilament RB-1 (Y304H, Ethicon, Somerville, NJ, USA) and 3-0 plain gut monofilament FS-2 (H822H, Ethicon, Somerville, NJ, USA). After the completion of the surgery, rats were transferred into the warm incubator with hourly subcutaneous injections of sterile 0.9% NaCl solution (10 mL/kg). For the next 2 days, animals received intraperitoneal injections of nefopam (10 mg/kg) and co-trimoxazole (50 mg/kg) twice-daily.

Four or twelve weeks postoperation, animals (*n* = 6 per group per time point) were euthanised by an intraperitoneal injection of a sodium pentobarbital (100 mg/kg body weight) with the subsequent excision of calvarial bones and adjacent tissues at the operation site. Excised tissues were then fixed in in two changes of 10% neutral phosphate buffered formalin (B06-003, BioVitrum, St. Petersburg, Russian Federation) for 48 h at 4 °C.

### 4.3. Complete Bblood Count and Biochemical Analysis

Complete blood count was measured and biochemical analysis was performed upon the rats were euthanised at 12 weeks postimplantation (*n* = 6 animals per group) utilising the automated analysers Hemalite 1280vet (Dixion, Moscow, Russian Federation) and A25 (Biosystems, Barcelona, Spain), respectively. In the complete blood count, we determined white blood cell count, red blood cell count, haemoglobin level, haematocrit, mean corpuscular volume, and platelet count. In a biochemical analysis, we measured total protein, albumin, globulins, glucose, triglycerides, cholesterol, urea, creatinine, bilirubin, uric acid, lactate dehydrogenase, alkaline phosphatase, alanine aminotransferase, and aspartate aminotransferase.

### 4.4. Microcomputed Tomography

To assess the volume and area of the repaired tissue and calculate the tissue and bone mineral density, formalin-fixed samples were visualised by means of microcomputed tomography (SkyScan 1172, Bruker, Billerica, MA, USA) at 8 µm voxel size after the 24-h rehydration in 0.9% NaCl solution. To maintain normal hydration during the scanning, bones were wrapped in the Parafilm M (P7543, Sigma-Aldrich, St. Louis, MO, USA). Calibration of bone mineral density was performed using Hounsfield units. For the calibration, we used HA samples with 8 mm diameter and 0.25 or 0.75 g/cm^3^ mineral density. In contrast to bone mineral density, tissue mineral density was defined as a density specific to the mineralised tissue and therefore excluding any surrounding non-bone tissue. Scanning was carried out with 0.75 mm aluminium filter to reduce beam hardening. Semiquantitative image analysis was performed utilising the CTAn software (Bruker, Billerica, MA, USA).

### 4.5. Histological Examination

Samples were placed in a decalcifying solution (06-004, BioVitrum, St. Petersburg, Russian Federation) for 4 days, dehydrated in 7 changes (5 h each) of isopropanol (06-002, BioVitrum, St. Petersburg, Russian Federation), impregnated in two changes (2 h each) of paraffin (Histomix Extra, 10342, BioVitrum, St. Petersburg, Russian Federation), and embedded into paraffin. Tissues were then sectioned in the centre and at the margins of the defect (5 sections across the sample, 5 µm thickness) and stained with hematoxylin and eosin (ab245880, Abcam, Cambridge, UK) according to the manufacturer’s protocol. Sections were evaluated by light microscopy (AxioImager.A1, Carl Zeiss, Oberkochen, Germany) in a blinded fashion to evaluate the extent of the repair.

### 4.6. Statistical Analysis

Statistical analysis was performed using GraphPad Prism 7 (GraphPad Software, San Diego, CA, USA). For descriptive statistics, data were represented by the mean and standard deviation. Time series data were analysed by a two-way analysis of variance with Tukey’s multiple comparisons test. Groups were compared by a one-way analysis of variance also with Tukey’s multiple comparisons test. *p* values ≤0.05 were regarded as statistically significant.

## 5. Patents

The implants used in this study were developed using the patent RU2665962C1 “Bioresorbable biological matrix for repairing bone tissue defects and method for the production thereof” and patent RU2693606C1 “Method for producing highly purified mineral matrix and use thereof”.

## Figures and Tables

**Figure 1 materials-13-03393-f001:**
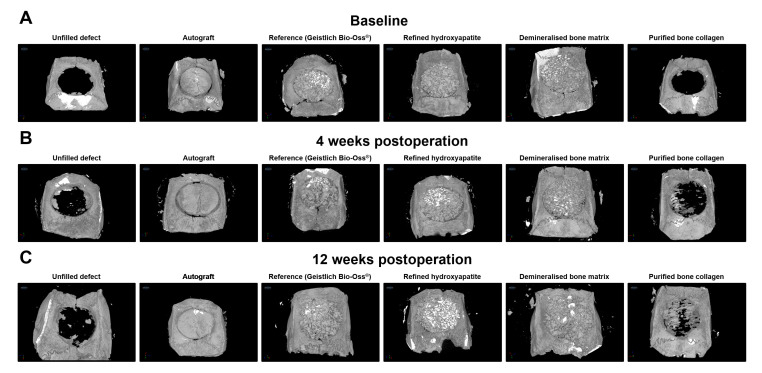
Representative microcomputed tomography images demonstrating filling of the critical-sized (8 mm diameter) rat calvarial defect upon the implantation of bovine-derived HA, DBM, and COLL at the baseline (**A**) and in a (**B**) short- and (**C**) long-term. Unfilled defect, autograft, and Geistlich Bio-Oss^®^ represent a negative control, positive control, and reference product, respectively. HA—refined hydroxyapatite powder, DBM—demineralised bone matrix powder, COLL—purified bone collagen membranes.

**Figure 2 materials-13-03393-f002:**
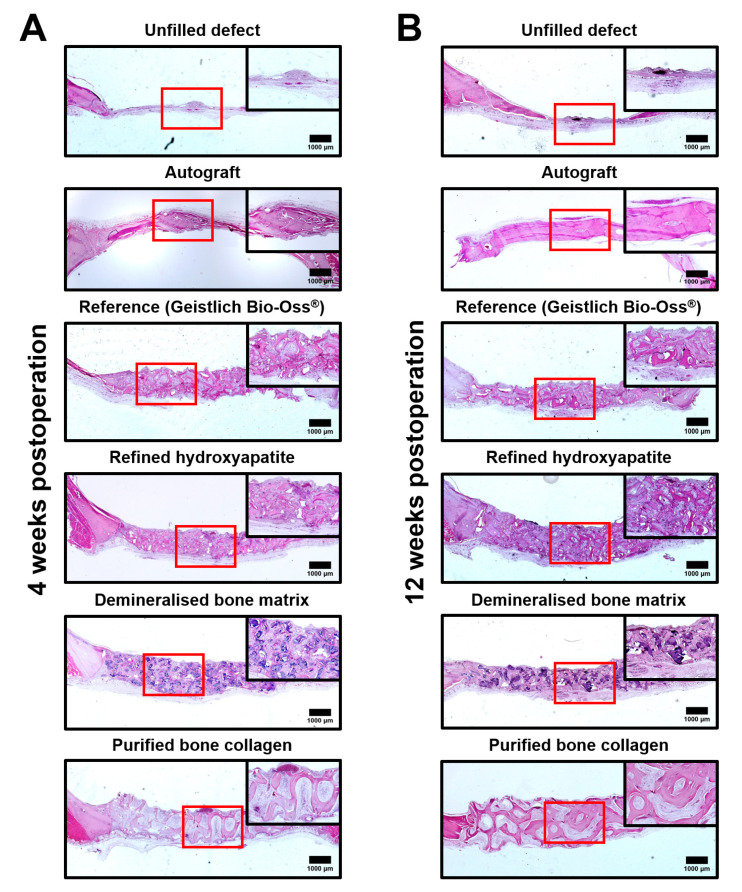
Representative hematoxylin and eosin-stained sections of rat calvarial bone tissues demonstrating filling of the critical-sized (8 mm diameter) rat calvarial defect upon the implantation of bovine-derived HA, DBM, and COLL in a (**A**) short- and (**B**) long-term. Close-ups (red squares demarcating the inserts) show areas with ongoing repair. Unfilled defect, autograft, and Geistlich Bio-Oss^®^ represent a negative control, positive control, and reference product, respectively. HA—refined hydroxyapatite powder, DBM—demineralised bone matrix powder, COLL—purified bone collagen membranes.

**Figure 3 materials-13-03393-f003:**
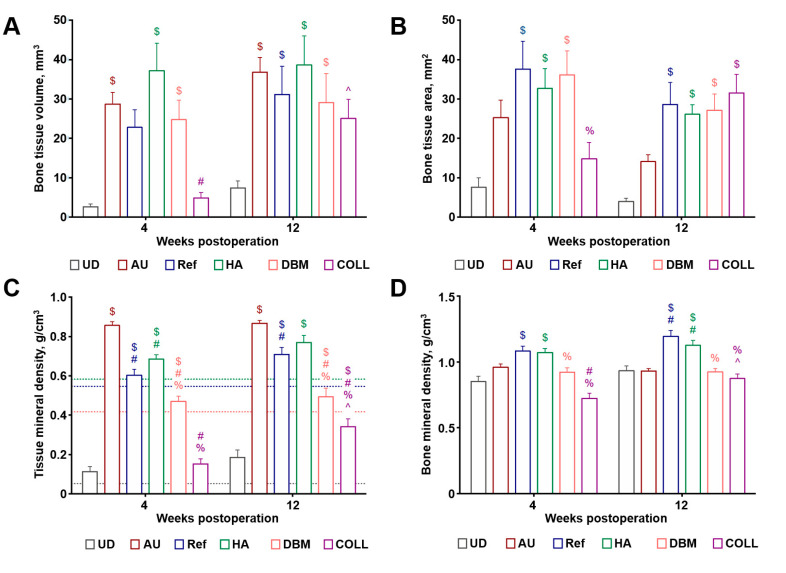
Evaluation of bone repair features upon the implantation of bovine-derived HA, DBM, and COLL into a critical-sized (8 mm diameter) rat calvarial defect in a short- and long-term. Microcomputed tomography measurements of (**A**) bone tissue volume; (**B**) bone tissue area; (**C**) tissue mineral density; and (**D**) bone mineral density. Unfilled defect, autograft, and Geistlich Bio-Oss^®^ represent a negative control, positive control, and reference product, respectively. *N* = 12 animals per group, one-way analysis of variance with Tukey’s multiple comparisons test. UD—unfilled defect, AU—autograft, Ref—reference product (Geistlich Bio-Oss^®^), HA—refined hydroxyapatite powder, DBM—demineralised bone matrix powder, COLL—purified bone collagen membranes. $ means statistically significant differences (*p* ≤ 0.05) compared with UD, # means statistically significant differences compared with AU; % means statistically significant differences compared with Ref; ^ means statistically significant differences compared with HA.

**Figure 4 materials-13-03393-f004:**
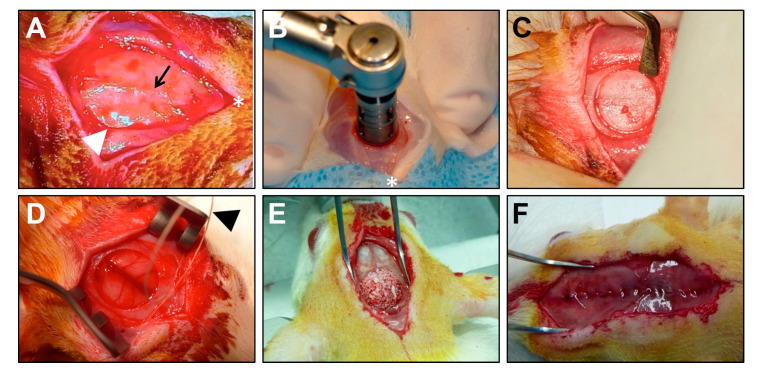
Representative images demonstrating the implantation of HA into the critical-sized (8 mm diameter) rat calvarial defect. (**A**) Access to the calvarial bones; (**B**) excision of the calvarial bones using a trephine; (**C**) calvarial bones ready to be excised; (**D**) calvarial defect upon the excision of the respective bone tissue; (**E**) filling of the calvarial defect with HA; and (**F**) closure of the periosteum over the implant. HA—hydroxyapatite.

**Table 1 materials-13-03393-t001:** Experimental design.

Experimental Group (*n* = 12 Per Each)	Sample Biomaterial
HA	Refined bovine hydroxyapatite powder
DBM	Demineralised bovine bone matrix powder
COLL	Purified bovine bone collagen membranes
Ref	Geistlich Bio-Oss^®^, powder
AU	Autograft (excised rat calvarial bones)
UD	None (unfilled defect)

HA—refined hydroxyapatite powder, DBM—demineralised bone matrix powder, COLL—purified bone collagen membranes, Ref—reference product (Geistlich Bio-Oss^®^), AU—autograft, UD—unfilled defect.

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
