# Peer review of "Native Bovine Hydroxyapatite Powder, Demineralised Bone Matrix Powder, and Purified Bone Collagen Membranes Are Efficient in Repair of Critical-Sized Rat Calvarial Defects"

_materials, 2020, doi:10.3390/ma13153393_

Round 1
Reviewer 1 Report
The paper titled “Native bovine hydroxyapatite powder, demineralised bone matrix powder, and purified bone collagen membranes are efficient in repair of critical-sized rat calvarial defects” is an interesting work that evaluated the efficacy of bone repair using various native bovine biomaterials (refined hydroxyapatite, demineralised bone matrix, and purified bone collagen) as compared with commercially available bone mineral and bone autografts.
Major comments:
- In section 2.2 , please add initial Microcomputed tomography scan at time t=0 (immediately after surgery). It will be much accurate to assess the healing of process of the defect.
- In figure 4 " Representative hematoxylin and eosin-stained sections of rat calvarial bone tissues" please add a picture , for each time point, with higher objective (for example x40) , than it will be possible to distinguish in the regeneration of the new bone (for example : osteocyte in the lacuna) .
Author Response
The paper titled “Native bovine hydroxyapatite powder, demineralised bone matrix powder, and purified bone collagen membranes are efficient in repair of critical-sized rat calvarial defects” is an interesting work that evaluated the efficacy of bone repair using various native bovine biomaterials (refined hydroxyapatite, demineralised bone matrix, and purified bone collagen) as compared with commercially available bone mineral and bone autografts.
Major comments:
1. In section 2.2, please add initial Microcomputed tomography scan at time t=0 (immediately after surgery). It will be much accurate to assess the healing of process of the defect.
We agree with the reviewer. We have added such images to the Figure 1 (former Figure 3 as former Figures 1 and 2 have been moved to the supplement in response to the comments of another reviewer).
2. In figure 4 "Representative hematoxylin and eosin-stained sections of rat calvarial bone tissues" please add a picture, for each time point, with higher objective (for example x40), than it will be possible to distinguish in the regeneration of the new bone (for example: osteocyte in the lacuna).
We have added close-ups of the areas with osteocytes in the lacuna and marked them within the main figure. Unfortunately, we currently do not have an access to our histological samples due to the manufacturer's expertise (they are evaluated by the medical drugs&device registration committee).
We sincerely thank the reviewer for the valuable comments.
Reviewer 2 Report
Dear editor,
In reference to the manuscript entitled "Native bovine hydroxyapatite powder, demineralised bone matrix powder, and purified bone collagen membranes are efficient in repair of critical-sized rat calvarial defects", I would like to send you my review comments.
In this manuscript, the authors compare the in vivo bone repair efficacy using several native bovine biomaterials. The manuscript is well designed, with a high scientific soundness and the quality of presentation is good. However, from my point of view the originality of this work is not very high, since authors are comparing several compounds that have already being individually tested. Moreover, experiments carried out are very simple and they should explore more ways to show the bone regeneration.
Next, you could find some specific comments:
- Table 1, experimental design. I don´t find it´s necessary to include some information of this table. If authors want to maintain it, I would remove the "sample collection" and "methods of examination" columns and I would remake it just to show the different materials they used (and acronyms?).
- Regarding to figures 1, 2 and table 2, I think they don´t bring us too much information to use as a main figure. These materials are already used and It´s widely known that they are not cytotoxic, so it´s normal to expect that they won´t produce any adverse reaction in the body. I suggest to include them as supplemental figures.
- I suggest to use acronyms for the different materials along the manuscript. For example, this title seems very large and it´s "difficult" to read it
"2.2. Native refined hydroxyapatite and demineralised bone matrix are highly and equally efficient for the repair of critical-sized rat calvarial defect while purified bone collagen demonstrates the highest bone regeneration rate"
->
NatHyd and DBM are highly and equally efficient for the repair of critical-sized rat calvarial defect while PBCol demonstrates the highest bone regeneration rate"
- 2.2. Micro CT:
"131 Microcomputed tomography examination showed complete repair of the defect upon its filling 132 with..."
From my point of view, you are not demonstrating with those images that bone is regenerated. You are using bone mineral and filling the space, it has the same density so maybe we are seeing just the material occupying the space. I would rephrase that sentence. Could you explain further the methods for bone assessment by μCT images?
- H-E images: Could authors include image magnifications in order to show how osteoblasts and other cells are penetrating into the biomaterial area? Some other stainings like Von Kossa or Masson´s Tricrhome would be recommended to make this section more robust.
Author Response
Dear editor,
In reference to the manuscript entitled "Native bovine hydroxyapatite powder, demineralised bone matrix powder, and purified bone collagen membranes are efficient in repair of critical-sized rat calvarial defects", I would like to send you my review comments.
In this manuscript, the authors compare the in vivo bone repair efficacy using several native bovine biomaterials. The manuscript is well designed, with a high scientific soundness and the quality of presentation is good. However, from my point of view the originality of this work is not very high, since authors are comparing several compounds that have already being individually tested. Moreover, experiments carried out are very simple and they should explore more ways to show the bone regeneration.
Next, you could find some specific comments:
Table 1, experimental design. I don´t find it´s necessary to include some information of this table. If authors want to maintain it, I would remove the "sample collection" and "methods of examination" columns and I would remake it just to show the different materials they used (and acronyms?).
We agree and reworked Table 1 as suggested by the reviewer.
Regarding to figures 1, 2 and table 2, I think they don´t bring us too much information to use as a main figure. These materials are already used and It´s widely known that they are not cytotoxic, so it´s normal to expect that they won´t produce any adverse reaction in the body. I suggest to include them as supplemental figures.
We agree and moved Figure 1, Figure 2, and Table 2 to the Supplement (in the revised manuscript they are designated as Supplementary Figure 1, Supplementary Figure 2 and Supplementary Table 1).
I suggest to use acronyms for the different materials along the manuscript. For example, this title seems very large and it´s "difficult" to read it
"2.2. Native refined hydroxyapatite and demineralised bone matrix are highly and equally efficient for the repair of critical-sized rat calvarial defect while purified bone collagen demonstrates the highest bone regeneration rate"
->
NatHyd and DBM are highly and equally efficient for the repair of critical-sized rat calvarial defect while PBCol demonstrates the highest bone regeneration rate"
We agree and used the following acronyms (to make the acronyms in the text consistent to those in the figures) in the revised manuscript:
HA for hydroxyapatite powder
DBM for demineralised bone matrix powder
COLL for purified bone collagen membranes
2.2. Micro CT:
"131 Microcomputed tomography examination showed complete repair of the defect upon its filling 132 with..."
From my point of view, you are not demonstrating with those images that bone is regenerated. You are using bone mineral and filling the space, it has the same density so maybe we are seeing just the material occupying the space. I would rephrase that sentence. Could you explain further the methods for bone assessment by μCT images?
We completely agree with this and rephrased the respective terminology throughout the manuscript, having also expanded the methods to analyze microCT images.
H-E images: Could authors include image magnifications in order to show how osteoblasts and other cells are penetrating into the biomaterial area? Some other stainings like Von Kossa or Masson´s Tricrhome would be recommended to make this section more robust.
We have added close-ups of the areas with osteocytes in the lacuna and marked them within the main figure. Unfortunately, we currently do not have an access to our histological samples due to the manufacturer's expertise (they are evaluated by the medical drugs&device registration committee).
We sincerely thank the reviewer for the valuable comments.
Reviewer 3 Report
Review for materials-869735-peer-review-v1
General Comments: Very nicely designed, written, and discussed. All of my comments below are minor, but a few general comments:
1) When writing bovine compounds (HA, DBM, collagen) indicate “bovine” before them throughout the manuscript. This will really help the reader since there are 6 groups being compared.
2) Table 1 is a very nice addition.
3) For Fig 1 and 2, at least indicate “NS” for non-significant. Table 2 is fine since the last column indicates the ANOVA P value.
More specific comments and text editing/text corrections and listed below by line number.
More Specific Comments:
Title – None
Abstract
1) Line 31 – Remove “the” before “bovine bone”
Introduction
1) Line 41 – Insert “a” before “high incidence”
Results – None
Discussion – None
Materials and Methods
1) The implant preparation section is very nice.
2) Line 273 – Indicate the gender/sex of the rats.
3) Line 352 – Indicate the post-test used with the 2-way ANOVA.
Figures, Tables, and Legends – See general comments above.
Author Response
General Comments: Very nicely designed, written, and discussed. All of my comments below are minor, but a few general comments:
1) When writing bovine compounds (HA, DBM, collagen) indicate “bovine” before them throughout the manuscript. This will really help the reader since there are 6 groups being compared.
We agree with the reviewer and indicated that HA, DBM and purified bone collagen were bovine-derived materials throughout the manuscript.
2) Table 1 is a very nice addition.
3) For Fig 1 and 2, at least indicate “NS” for non-significant. Table 2 is fine since the last column indicates the ANOVA P value.
We indicated NS to underline the absence of statistically significant differences in the Supplementary Figure 1 and Supplementary Figure 2 (former Figures 1 and 2 as they have been moved to the supplement in response to the comments of another reviewer).
More specific comments and text editing/text corrections and listed below by line number.
More Specific Comments:
Title – None
Abstract
1) Line 31 – Remove “the” before “bovine bone”
We corrected.
Introduction
1) Line 41 – Insert “a” before “high incidence”
We corrected.
Results – None
Discussion – None
Materials and Methods
1) The implant preparation section is very nice.
2) Line 273 – Indicate the gender/sex of the rats.
We indicated that all rats were of male gender.
3) Line 352 – Indicate the post-test used with the 2-way ANOVA.
We have added this information (Tukey’s multiple comparisons test).
Figures, Tables, and Legends – See general comments above.
We sincerely thank the reviewer for the valuable comments.
Round 2
Reviewer 1 Report
No comments. The article was corrected according to my comments.
Author Response
No comments. The article was corrected according to my comments.
We sincerely thank the reviewer for the constructive criticism of our paper.
Reviewer 2 Report
Dear editor,
Authors have implemented most of the changes suggested by reviewers.
As a final comment, I suggest to put the scale bars in figure 2 in the same place in all the pictures. It would be great if you put it in a corner. Moreover, you can just draw the scale bar in the pictures and specify the size in the footnote (i.e. "scale bar: 1000 um). It would be more elegant.
After that, I would accept the paper.
Author Response
Authors have implemented most of the changes suggested by reviewers.
As a final comment, I suggest to put the scale bars in figure 2 in the same place in all the pictures. It would be great if you put it in a corner. Moreover, you can just draw the scale bar in the pictures and specify the size in the footnote (i.e. "scale bar: 1000 um). It would be more elegant.
After that, I would accept the paper.
We agree with the reviewer and have drawn the scale bar at the bottom right corner in all the pictures within the Figure 2.